# Machine learning analysis of TCGA cancer data

Jose Liñares-Blanco[1,2], Alejandro Pazos[1,2,3] and
Carlos Fernandez-Lozano[1,2,3]

[1] CITIC-Research Center of Information and Communication Technologies, University of
A Coruna, A Coruña, Spain
[2] Department of Computer Science and Information Technologies, Faculty of Computer Science,
University of A Coruna, A Coruña, Spain
[3] Grupo de Redes de Neuronas Artificiales y Sistemas Adaptativos. Imagen Médica y Diagnóstico
Radiológico (RNASA-IMEDIR). Complexo Hospitalario Universitario de A Coruña (CHUAC),
SERGAS, Universidade da Coruña, Instituto de Investigación Biomédica de A Coruña (INIBIC),
A Coruña, Spain



## ABSTRACT

In recent years, machine learning (ML) researchers have changed their focus towards
biological problems that are difficult to analyse with standard approaches. Large
initiatives such as The Cancer Genome Atlas (TCGA) have allowed the use of omic
data for the training of these algorithms. In order to study the state of the art, this
review is provided to cover the main works that have used ML with TCGA data.
Firstly, the principal discoveries made by the TCGA consortium are presented. Once
these bases have been established, we begin with the main objective of this study, the
identification and discussion of those works that have used the TCGA data for the
training of different ML approaches. After a review of more than 100 different
papers, it has been possible to make a classification according to following three
pillars: the type of tumour, the type of algorithm and the predicted biological
problem. One of the conclusions drawn in this work shows a high density of studies
based on two major algorithms: Random Forest and Support Vector Machines.
We also observe the rise in the use of deep artificial neural networks. It is worth
emphasizing, the increase of integrative models of multi-omic data analysis.
The different biological conditions are a consequence of molecular homeostasis,
driven by both protein coding regions, regulatory elements and the surrounding
environment. It is notable that a large number of works make use of genetic
expression data, which has been found to be the preferred method by researchers
when training the different models. The biological problems addressed have been
classified into five types: prognosis prediction, tumour subtypes, microsatellite
instability (MSI), immunological aspects and certain pathways of interest. A clear
trend was detected in the prediction of these conditions according to the type of
tumour. That is the reason for which a greater number of works have focused on the
BRCA cohort, while specific works for survival, for example, were centred on the
GBM cohort, due to its large number of events. Throughout this review, it will be
possible to go in depth into the works and the methodologies used to study TCGA
cancer data. Finally, it is intended that this work will serve as a basis for future
research in this field of study.

Corresponding author
Carlos Fernandez-Lozano,
carlos.fernandez@udc.es

## INTRODUCTION

The appearance of the carcinogenic phenotype is the consequence of an alteration of one or more genes. In addition, the appearance of subtypes occurs in different ways in individuals of a population. Hence, a major problem that arises in cancer is the difficulty in its genetic diagnosis. Similar to Mendelian diseases, where the disease develops due to the alteration in the function of a single gene, the development of cancer is a consequence of epistatic behaviour of genes. There is already an extremely large search space in the identification of alterations in a single gene, including exonic and intronic mutations, single nucleotide polymorphisms (SNPs), copy number variants, indels, post-transcriptional alterations, post-translational alterations, three-dimensional assembly of the protein, epigenetic modifications, etc. Thus, the search space for alterations when we encounter a subgroup of 40 genes is immense. When we do not know exactly which genes are involved, we have to search among the more than 20,000 coding regions or even in whole genome sequence. In these cases the search space grows to incalculable levels. All this complexity is the result of intermolecular communications in and among cells, a phenomenon that constitutes an environment of molecular communication that is extremely complicated to understand and identify.

In order to lay the foundation and achieve great advances in the prevention, early detection, stratification and success in the treatment of cancer, it is necessary to identify the complete changes generated by each type of cancer in its genome. Further, researchers must understand how these changes interact with the cancer microenvironment, intra- and intercellularly, to manifest itself. Hence, the National Cancer Institute (NCI) and the National Human Genome Research Institute (NHGRI) of the United States established The Cancer Genome Atlas (TCGA), with the aim of obtaining comprehensive multidimensional genomic maps of all key changes in several types and subtypes of cancer (*The Cancer Genome Atlas Research Network, 2008*). An initial pilot project in 2006 confirmed that an atlas of these changes could be specifically created for different types of cancer. Subsequently, TCGA has collected tissues from more than 11,000 cancer and healthy patients, an endeavour that allows the study of more than 33 types and subtypes of cancer, including 10 rare cancers. The most interesting aspect of this initiative is that all the information is free and accessible to any researcher who wants to focus their efforts on the disease. The different types of data presented by the TCGA project are summarised in Table 1 and Fig. 1 shows, for each cancer type, the percentage that each data type represents in the subtype's total. Data are provided open access to the community, a factor that facilitates the generation of novel models without requiring an initial financial investment to obtain the data. Therefore, there are increasingly specific models for the analysis of omics data. In particular, the rise and success of machine learning (ML) techniques to process a large amount of data is revolutionising bioinformatics and

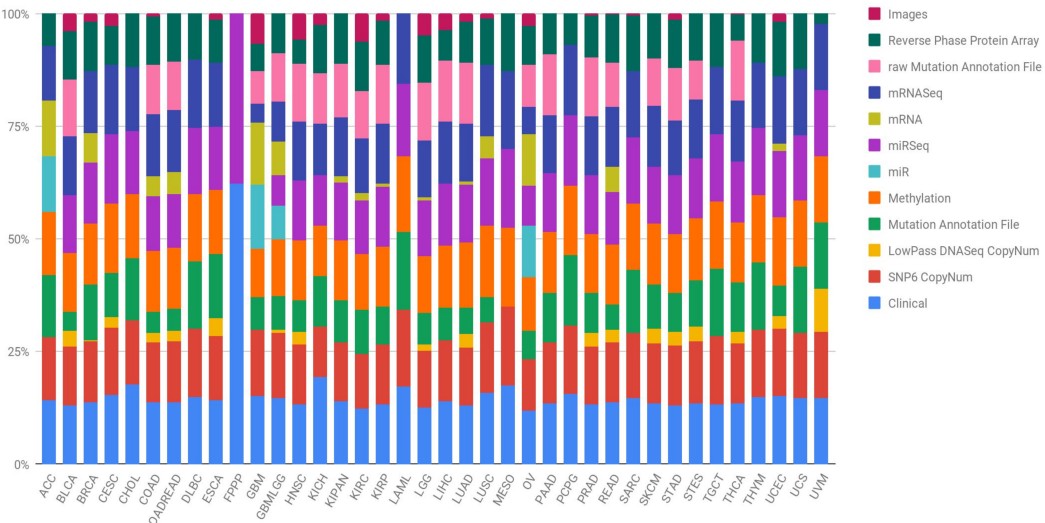

**Figure 1 Quantification of the number of samples in the TCGA repository, classified by type of tumour and type of biotechnological analysis.** Clin, Clinical; SNP6, SNP6 CopyNum; DNAseq, Low-Pass DNASeq CopyNum; Mutat, Mutation Annotation File; Met, Methylation; rawMut, rawMutation Annotation File; Prot, Reverse Phase Protein Array.

conventional forms of genetic diagnosis. These methods have focused on making predictions by using general learning algorithms to find patterns in complex, larger and hard-to-handle problems. In addition, these ML methods work really well with very large datasets, even when the number of variables in each observation is much greater than the total number of observations ($n << p$).

This survey presents the state-of-the-art research on TCGA analysis using machine learning. Efforts have involved both supervised and unsupervised learning problems, as well as survival analysis, disease prognosis, cancer staging and pathways analysis to analyse different types of data ranging from multi-omics human cancer data to imaging. Therefore, review articles are needed to show an overview of machine learning-based analysis of TCGA data to highlight the findings and to discuss future research lines so that the obtained knowledge is useful and can be translated to clinical practice.

There are few published review articles on machine learning for biomedical genomic analysis (*Leung et al., 2015*; *Karczewski & Snyder, 2018*). These review articles are before 2018 and do not present a discussion on TCGA data nor a discussion on machine learning results neither present a multi-omic and imaging point of view for different biological questions. To the best of our knowledge, no survey has been conducted on Machine Learning analysis of TCGA using multi-level cancer data. Thus, this survey aims to present a comprehensive summary of the previous machine learning approaches applied to TCGA during the span of 2008-2020. The contributions of this review are:

- This review includes exhaustive review of the main results obtained by the TCGA consortium using conventional approaches in order to understand if machine learning is increasing the knowledge in the area.
- This review includes machine learning results by the TCGA consortium.

**Table 1 Different types of data present in the TCGA repository.**

| DNA Sequencing | Whole genome sequences |
|---|---|
| | Whole exome sequences |
| | Sequences traces |
| | Mutations, including coding, splice site, germline and noncoding somatic variants |
| RNA sequencing | mRNA sequences (calculated expression per gene, exon, splice junction and isoform) |
| | miRNA sequences (calculated expression per miNRA and isoform) |
| | Total RNA sequences (calculated expression per gene, exon, splice junction and isoform) |
| | Expression signals per gene, exon, splice junction, miRNA and isoform |
| Copy number | Arrays (raw, unnormalized, normalized) |
| | Low-pass DNA sequencing (whole genomes sequences, variants and coverage) |
| Array-based expression | Gene expression (raw, normalized and calls) |
| | Exon expression (raw, normalized and calls) |
| | miRNA expression (raw, normalized and calls) |
| DNA methylation | Array-based methylation (raw signal intensity, calculated beta values) |
| Other | Protein expression (high-resolution images of protein arrays, raw signals, normalized expression and mass spectrometry protein) |
| | Microsatelite instability (markers and classification) |
| | ATAC-seq (chromatine accesibility) |
| Metadata | Clinical information about patients (e.g., sex, race, ethnicity, drugs taken, metastasis status and response to treatment) |
| | Information about samples (e.g., the weight of a sample portion, days to collect and time of freezing) |
| | Images of the tumors |

- A classification of supervised, unsupervised and clustering methods that may point researchers to new approaches or new problems.
- Identification of data types mostly used in machine learning research of TCGA.
- A comprehensive discussion on biological questions solved by machine learning algorithms: prognosis, immunological phenotype, pathways, MSI status, and subtype prediction.
- A deeper examination of the most used TCGA cohort: Breast Cancer Adenocarcinoma (BRCA).
- We point data integration approaches as the future trend in TCGA analysis using machine learning.

We believe that researchers in machine learning, bioinformatics, biology, computational biology and data integration would benefit from the findings of this exhaustive and comprehensive review.

This manuscript is organised as follows. "Survey Methodology" explains the methodology used in this survey. "TCGA Consortium" presents the main results obtained by the TCGA consortium. In "Machine Learning as a Source of New Knowledge", we review the TCGA efforts with those algorithms as well as we present the most used

algorithms on supervised, unsupervised and clustering approaches for external researchers. Special attention with a subsection on medical imaging analysis using deep learning approaches in recent years. "Biological Questions Solved by Machine Learning Algorithms" discusses the capability of those algorithms to solve the biological problem with the highest performance score and find that the predictions are biologically of relevance. To this aim we divide and study five biological problems: prognosis, immunological phenotype, pathways, MSI and subtypes prediction. We finish with special emphasis on the analysis of the BRCA cohort. Finally, we conclude the review in 'Conclusions'.

## SURVEY METHODOLOGY

This work is based on a literature review in machine learning-based analysis of TCGA cancer data. We searched for the main findings of the TCGA consortium using classical statistical approaches and works using machine learning and classify them into supervised, unsupervised and clustering methods. Furthermore, we considered of relevance to answer to the intitial biological question with sense, not only with a higher performance score. The search keywords, data sources and on criteria are discussed.

### Search keywords

We initially reviewed the original TCGA consortim publication in order to carefully select the search keywords. The keywords used for the survey included the following terms to find the relevant papers: 'machine learning', 'TCGA'. We used the 'AND' and 'OR' Boolean operators to combine terms. After the initial subset of papers we refined the search keywords according with the most used machine learning models, type of problems and biological question: 'clustering', 'computer vision', 'deep learning', 'random forest', 'support vector machines', 'linear model', 'survival', 'MSI', 'prognosis', 'pathway', 'subtypes' or 'phenotype'.

### Data sources

The papers included in this survey were retrieved from prominent journals indexed in diverse quality databases: Pubmed and Scopus.

### Article inclusion/exclusion criteria

We decided which articles are eligible for the survey under the following inclusion/exclusion criteria:

- Inclusion criteria:

    - manuscripts written in the English language and published by indexed journals in Pubmed to ensure the health science specialization and Scopus using TCGA as the main source of data

- Exclusion criteria:

    - manuscripts using machine learning marginally or without solid biological conclusions

    - manuscripts in preprint without peer review

## Article selection

The TCGA consortium papers were identified in the website and were included. Initially 345 papers were identified in Pubmed and Scopus using the search keywords. Of these, we filtered by the inclusion/exclusion criteria. In addition, duplicated papers retrieved from multiple sources were removed. Finally, more than 150 articles were included.

## TCGA CONSORTIUM

TCGA began as a pilot project for 3 years, with a focus on the characterisation of three types of human cancer: glioblastoma multiforme (GBM), lung squamous cell carcinoma (LUSC) and ovarian cancer (OV). TCGA currently presents data from a total of 38 different cohorts. Four of them (COADREAD, GBMLGG, KIPAN and STES) are not original—they are combinations of other cohorts. Among the remaining 34 cancer cohorts are tumours of different tissue types, as can be seen in Table 2. To date, TCGA has characterised and published about 33 different types of tumours in leading international journals. Table 2 provides greater depth for each of the publications that TCGA has made in each recruited cohort.

In 2018, a series of works were published in Cell editorial, where they were exhaustively analysed the samples recruited throughout the project. These studies led to the identification and examination of mechanisms that underlie all types of tumours. These findings allow researchers to draw conclusions about tumour origins, molecular biology and subtyping. In this series of publications—and in order to understand the molecular biology underlying cancer—the TCGA consortium cross-checked general molecular aspects in all tumour types. To this end, they exhaustively studied, in the more than 10,000 samples stored in their repository, the process of alternative splicing (*Kahles et al., 2018*) and they identified the specific variants (*Huang et al., 2018*) and driver genes (*Bailey et al., 2018*) that generate greater predisposition to tumour development. They also analysed the effect of enhancer activation on different tumour types (*Chen et al., 2018a*) and the effect of aneuploidy (*Taylor et al., 2018*). They also catalogued the variants of the 10 pathways that are most frequently altered in most tumours (*Sanchez-Vega et al., 2018*), in addition to alterations in genes related to the ubiquitin (*Ge et al., 2018*), DNA damage repair (*Knijnenburg et al., 2018*) and the MYC pathways (*Schaub et al., 2018*).

The consortium also features a strong technology component; they published an integrated pancancerous clinical data resource from TCGA with the aim of driving the analysis of high-quality survival results (*Liu et al., 2018a*). In addition, they conducted studies where they used ML and deep learning algorithms to identify stemness features in tumour cells (*Malta et al., 2018*), the prediction of Ras pathway activation (*Way et al., 2018*) and the detection of tumour infiltrating lymphocytes using images (*Saltz et al., 2018*). In *Ellrott et al. (2018)* they described the Multi-Center Mutation Calling project, which aims to generate a complete encyclopaedia of somatic mutations from TCGA data that allows a robust analysis for different tumour types. They performed different studies that proposed new classifications among tumours. For example, they identified new immune tumour types across the 33 types of cancer that differ by somatic aberrations,

**Table 2 Enumeration of the different cohorts presented by the TCGA repository, classified according to the tissue of origin of the tumour.** In addition, the original paper published by the TCGA consortium is cited.

| Cancer type | Acronym | Tissue | Citation |
|---|---|---|---|
| Breast Ductal/Lobular Carcinoma | BRCA | Breast | (*The Cancer Genome Atlas Network, 2012b*; *Ciriello et al., 2015*) |
| Glioblastoma Multiforme | GBM | Central Nervous System | (*The Cancer Genome Atlas Research Network, 2008*; *Verhaak et al., 2010*; *Noushmehr et al., 2010*; *Brennan et al., 2013*; *The Cancer Genome Atlas Research Network, 2015*, *Ceccarelli et al., 2016*) |
| Lower Grade Glioma | LGG | Central Nervous System | (*The Cancer Genome Atlas Research Network, 2015*) |
| Adrenocortical Carcinoma | ACC | Endocrine | (*Zheng et al., 2016*) |
| Papillary Thyroid Carcionma | THCA | Endocrine | (*Agrawal et al., 2014*) |
| Paraganglioma & Pheochromocytoma | PCPG | Endocrine | (*Fishbein et al., 2017*) |
| Cholangiocarcinoma | CHOL | Gastrointestinal | (*Farshidfar et al., 2017*) |
| Colon Adenocarcinoma | COAD | Gastrointestinal | (*The Cancer Genome Atlas Network, 2012a*) |
| Rectal Adenocarcinoma | READ | Gastrointestinal | (*The Cancer Genome Atlas Network, 2012a*) |
| Esophageal Cancer | ESCA | Gastrointestinal | (*The Cancer Genome Atlas Research Network, 2017c*) |
| Liver Hepatocellular Carcionoma | LIHC | Gastrointestinal | (*Ally et al., 2017*) |
| Pancreatic Ductal Adenocarcinoma | PAAD | Gastrointestinal | (*Raphael et al., 2017*) |
| Stomach Cancer | STAD | Gastrointestinal | (*The Cancer Genome Atlas Research Network, 2014a*) |
| Cervical Cancer | CESC | Gynecologic | (*The Cancer Genome Atlas Research Network, 2017b*) |
| Ovarian Serous Cystadenocarcinoma | OV | Gynecologic | (*The Cancer Genome Atlas Research Network, 2011*) |
| Uterine Carcinosarcoma | UCS | Gynecologic | (*Cherniack et al., 2017*) |
| Uterine Corpus Endometrial Carcinoma | UCEC | Gynecologic | (*Levine, 2013*) |
| Head and Neck Squamous Cell Carcinoma | HNSC | Head and Neck | (*The Cancer Genome Atlas Network, 2015*) |
| Uveal Melanoma | UVM | Head and Neck | (*Robertson et al., 2017b*) |
| Acute Myeloid Leukemia | AML | Hematologic | (*The Cancer Genome Atlas Research Network, 2013*) |
| Thymoma | THYM | Hematologic | (*Radovich et al., 2018*) |
| Cutaneous Melanoma | SKCM | Skin | (*Akbani et al., 2015*) |
| Sarcoma | SARC | Soft Tissue | (*The Cancer Genome Atlas Research Network, 2017a*) |
| Lung Adenocarcinoma | LUAD | Thoracic | (*The Cancer Genome Atlas Research Network, 2014c*; *Campbell et al., 2016*) |
| Lung Squamous Cell Carcinoma | LUSC | Thoracic | (*The Cancer Genome Atlas Research Network, 2012c*; *Campbell et al., 2016*) |
| Mesothelioma | MESO | Thoracic | (*Hmeljak et al., 2018*) |
| Chromophobe Renal Cell Carcinoma | KICH | Urologic | (*Davis et al., 2014*) |
| Clear Cell Kidney Carcinoma | KIRC | Urologic | (*The Cancer Genome Atlas Research Network, 2013*) |

(Continued)

| Table 2 (continued) | | | |
|---|---|---|---|
| Cancer type | Acronym | Tissue | Citation |
| Papillary Kidney Carcinoma | KIRP | Urologic | (*The Cancer Genome Atlas Research Network, 2016*) |
| Prostate Adenocarcinoma | PRAD | Urologic | (*Abeshouse et al., 2015*) |
| Testicular Germ Cell Cancer | TGCT | Urologic | (*Shen et al., 2018*) |
| Urothelial Bladder Carcinoma | BLCA | Urologic | (*The Cancer Genome Atlas Research Network, 2014b*; *Robertson et al., 2017a*) |
| Diffuse Large B-cell Lymphoma | DLBC | Lymphatic tissue | |

microenvironment and survival (*Thorsson et al., 2018*). Furthermore, they classified tumours based on metabolic expression and subsequently proposed different subtypes that were not previously contemplated (*Peng et al., 2018*). In addition, they carried out exhaustive studies on groupings of tumours according to their origin in order to elucidate new therapeutic targets that might be useful for gastrointestinal adenocarcinomas (*Liu et al., 2018b*), gynaecological tumours and breast cancers (*Berger et al., 2018*) and squamous carcinomas (*Campbell et al., 2018*). In these papers, they performed clustering techniques to subtype patients into new groups for treatment or diagnosis. Finally, they studied tumours by cell (*Hoadley et al., 2018*) and tissue (*Hoadley et al., 2014*) of origins.

There are many results reported by TCGA that have had a very important impact on oncology. The results obtained by the consortium show a roadmap to follow and open countless avenues in this field where new research groups, until now unable to carry out their research globally, will be able to report important results in this field.

## MACHINE LEARNING AS A SOURCE OF NEW KNOWLEDGE

ML is the process by which machines acquire the ability to learn an action or behaviour. These processes are defined by different algorithms that enable the computer to learn a behaviour (classify, identify, etc.) and extract patterns from the data. These patterns are ultimately inherent knowledge of the problem to be analysed that the algorithms can extract and learn to identify. Subsequently, given a new case, these techniques can evaluate and predict to which group it is most likely to belong, always in accordance with prior knowledge. It is therefore critical that such techniques are applied with careful experimental design (*Fernandez-Lozano et al., 2016*) and that the data are as accurate as possible to define the problem. These techniques will learn and maximally exploit the intrinsic knowledge that underlies the data.

Depending on how this information extraction process is performed, we can speak of different approaches: supervised and unsupervised learning. Although in practice there are more types of learning, we will only focus on these two, mainly because these approaches have been the most widely used in biomedicine.

## The TCGA consortium and ML

The TCGA consortium has analysed cancer based on ML algorithms, sometimes with novel approaches specifically designed for the TCGA data. TCGA researchers recently presented a new ML that can predict the differentiation of certain tumour tissues (*Malta et al., 2018*). In this case, using data from non-differentiated stem cells and their differentiated progenitors (data obtained from public repositories), they constructed two classes of indicators that reflect epigenetic and genetic expression traits of the cells. Once they constructed these descriptors, they used a variant of one-class logistic regression to classify the different TCGA samples according to their degree of differentiation, a crucial characteristic for the development of the tumour and its invasive potential.

Another study (*Way et al., 2018*) used three types of omics platforms (expression, copy number and mutation) to predict the activation of the Ras pathway, which has been widely studied throughout oncological research. This model predicted whether this pathway was activated using RNAseq expression data. From the copy number and mutation data, the researchers were able to label the patients to design a supervised learning problem. Therefore, it was observed that certain omic patterns could be predicted from different omic data. This enables the prediction of a significant number of characteristics in tumours. This approach was also performed in another study by modifying the target in order to predict the activation of the TP53 pathway (*Knijnenburg et al., 2018*).

In other study, deep learning based on convolutional neural networks (CNN) mapped tumour infiltrating lymphocytes (TIL) based on haematoxylin and eosin (H&E) images. In this case, 13 types of TCGA tumours exhibited almost perfect performance when differentiating these cell types (*Saltz et al., 2018*). In this work, the TCGA consortium highlighted the importance of the images it stores and questions their relatively limited use by different researchers in comparison with omics platforms. The images in the TCGA repository will be discussed in the following sections.

## Popular ML models with TCGA data

The TCGA consortium has relied on both supervised and unsupervised ML techniques to extract new knowledge from its data. However, it is interesting to identify the work developed by other researchers who have used TCGA data. The approaches taken and the results obtained from the various published works will be discussed below.

Figure 2 shows the proportion of published papers according to the type of algorithm and the type of omic data used. We reviewed more than 100 papers that have used ML approaches with TCGA data. For each one, we identified: the algorithm and data type/s. Almost half of the identified works used variants of the support vector machine (SVM) or tree-based algorithms, followed by linear models as can be seen in Fig. 2A. On the other hand, Fig. 2B clearly shows that gene expression data is most abundant data type used in ML research. Other data types such as images, methylation, miRNA and copy number have been used, but majority in a combination with gene expression data.

The findings of this review highlight the low variability of reported research and analytical methods. It is true that the mostly used algorithms, Random Forest (RF) and SVM, as well as the types of omic data (expression) have reported promising results in

A

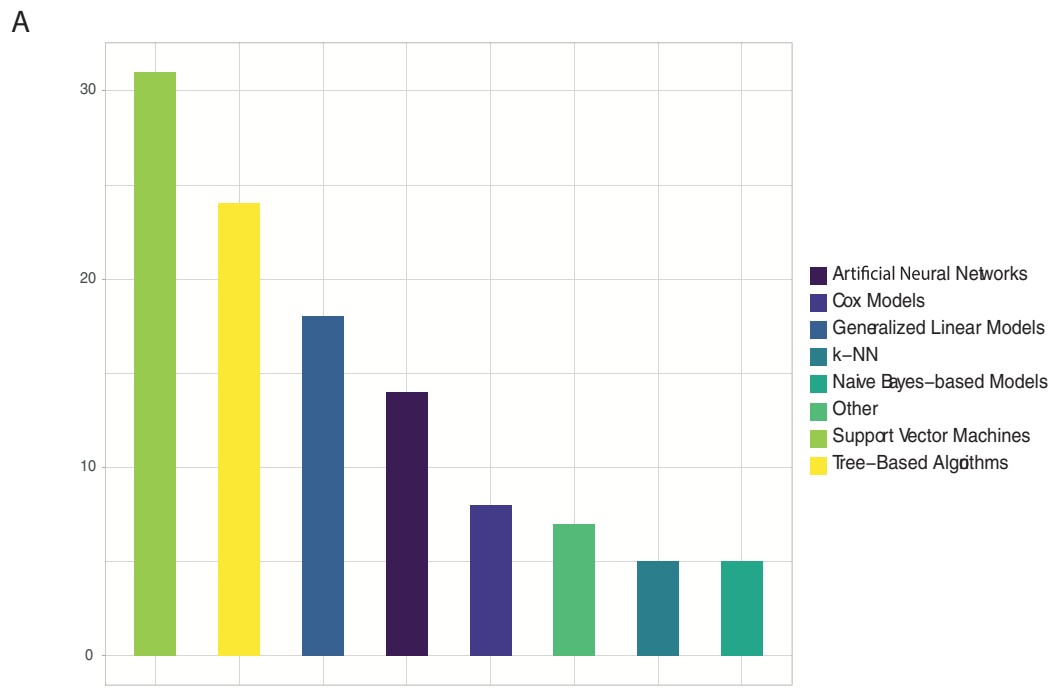

B

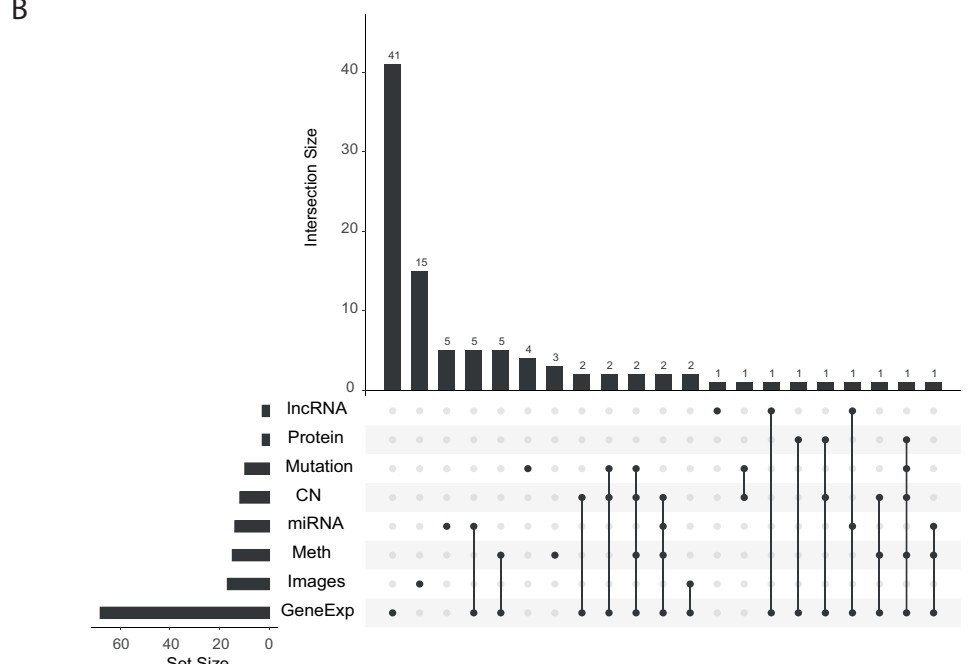

**Figure 2 (A) Number of papers that used each type of algorithm, and (B) relations between omics data used in each work.**

the biomedical field during the last years. We believe that the low variability in the approaches established by researchers is mainly due to two reasons. First, the intrinsic characteristics of biomedical data, and specifically the omic data, present a much greater number of characteristics than observations. This fact is generally not idyllic for the

training of ML algorithms. In this sense, the use of algorithms is mainly determined by the use of which type of omic data is being analyzed. In the context mentioned above, certain algorithms are able to handle some characteristics of the data better than others. For example, neural networks are more sensitive to the lack of observations than in this case RF, SVM or linear models. Given that the vast majority of works identified have used expression data, it is logical to observe a high density of works that have used RF or SVM type algorithms. On the other hand, those works that have used image data are more likely to use neuron networks. Secondly, there is no doubt that the possibilities in the exploitation of these data by ML algorithms are yet to be discovered. A break in the arrival of ML-based applications in the field of biomedicine has been detected. This is partly due to the complexity of the omic data, and the need for specialists in this field for its modelling and good practical use. Possible applications that could revolutionize the field of biomedicine could be the use of NLP (Natural Language Processing) algorithms for the analysis of Whole Genome Sequencing (WGS) data.

After all, if there is something to highlight in the results observed in the Fig. 2B is the trend towards more and more work integrating different omic data. Even so, this trend is not reflected in Fig. 2A, in which a variety of algorithms and/or new known and standardized methodologies that can solve this problem are not observed. This is the great challenge in the coming years presented by biomedicine, which could generate very useful predictions for tackling complex diseases, such as cancer.

### A general perspective of unsupervised learning with TCGA data

In oncology, clustering methods are extremely useful for subtyping or reclassification of patients in a particular cohort. Over the years, the classic clustering methods have been most widely used, including partitioning clustering or hierarchical clustering. Even today, they are widely used with their respective variations. For example, the TCGA consortium has used them to subtype different tumours (see publications in Table 2). The problem with these algorithms is that they can only model a single set of data and the concatenation of different types of data does not perform adequately. The complexity of the tumour is manifested at distinct biological levels; hence, methods that can accept different types of data are preferable. Thus, researchers developed a new integrative clustering method based on a joint latent variable model (iCluster) (*Shen, Olshen & Ladanyi, 2009*) and used it with TCGA data (*Shen et al., 2012*). *iCluster* fits a regularized latent variable model based clustering that generates an integrated cluster assigment based on joint inference across data types. In addition, the implementation in several programming languages is very intuitive. On other hand, an extended version (*iClusterPlus*) was also developed (*Mo et al., 2013*). One of the most important works using this method was (*Curtis et al., 2012*), identifying 12 different breast tumour subtypes.

In addition to beforementioned works, there are a huge examples of iCluster use with TCGA. For instance, in *Xie et al. (2019)* an integrative analysis was carried out with iCluster through RNAseq and proteomics data to analyse the OV subtype. The results showed two clusters with different survival rates; the method identified 18 mRNAs and 38 proteins as distinct molecules among subtypes. Another study proposed a modified

iCluster model to discover key processes in the tumour collection through unsupervised integration of multiple types of molecular data and functional annotations (*Bismeijer, Canisius & Wessels, 2018*). Further, *Mo et al. (2017)* described a novel modification (iClusterBayes) capable of jointly modelling omics data of continuous and discrete data types for the identification of tumour subtypes and relevant omics characteristics. In the work of *Kim et al. (2017)*, they modified this procedure to subtype patients using sequential double regularisation. Another pathway-based variant incorporates pathway data to group patients into cancer subtypes (*Mallavarapu et al., 2019*). Additionally, in *Jean-Quartier et al. (2021)* clustered GBM patients into several age subgroups with different age-related biomarkers. Finally, a work developed in *Nguyen et al. (2017)*, named PINS, allows omics data integration and molecular patient stratification automatically.

With the above, the trend in genome research is evident. An increasing number of works are attempting to integrate the greater amount of information provided by the different omic data into their models. Due to the complexity of cancer, stratifying patients according to a single source of information is becoming obsolete. Therefore, it is vitally important to improve models that are capable of multi-omic integration, as is the case with iCluster. Moreover, there is a need of novel approaches to automated medical decision pipelines building on machine learning, information fusion and explainability (*Holzinger et al., 2021*; *Barredo Arrieta et al., 2020*).

### Medical imaging as a data source for ML algorithms

An important event occurred in 2012 during the celebration of the ImageNet Large-Scale Visual Recognition Challenge (ILSVRC) (*Russakovsky et al., 2015*). A deep learning model (specifically, a CNN) halved the second best error rate in the image classification task. The goal of this challenge was the detection of objects and the classification of images using a large-scale database. Furthermore, deep learning algorithms can automatically find the best subset of features that describe the nuances of images. In addition, transfer learning was borne: an attempt to reuse the representation of the learning characteristic of one problem to solve another.

Deep learning techniques are on the rise in cancer research, namely for object detection and image classification. Initiatives such as TCGA offer the possibility of training deep learning models by making a large quantity of biomedical images available for research. Specifically, TCGA provides two types of images: tissue slide and digital imaging and communications in medicine (DICOM) images. DICOM images such as X-rays or computed tomography (CT), are used to extract quantitative characteristics from the images. Algorithms are trained to identify those characteristics. Histopathological images are used for direct image processing.

As discussed in previous sections, the TCGA consortium has used deep learning methods (*Saltz et al., 2018*). Specifically, they used CNN to detect tumour-infiltrating lymphocytes (TILs) based on H&E images in 13 tumour types. They reported a local spatial structure in the TIL patterns and their correlation with overall survival. These data modify densities and spatial structure among tumour types, immune subtypes and

molecular tumour subtypes. Spatial infiltration of lymphocytes might reflect particular aberration states of tumour cells.

Based on these findings, several studies have used this and other repositories to create their own models. It is important to distinguish among data types. On the one hand, there are works that have used radiological images for the classification of stages of gliomas (*Park et al., 2019*). In this work, they did not use the radiological images directly; rather, they extracted 250 characteristics from them to train their models, obtaining an area under the receiver operating characteristic curve (AUROC) of 72%. Notably, this model, which was validated with very heterogeneous cohorts such as TCGA, considerably reduced the performance. These results indicate that manual extraction of characteristics does not provide sufficient generalisation.

*Sun et al. (2018b)* utilised contrast-enhanced CT images and RNASeq data to assess CD8 cell infiltration in tumour biopsies. They first extracted features from both types of data to ultimately keep eight features and train an elastic-net regularised regression method. They used this signature to predict the response to anti-programmed cell death protein 1 (PD1) or anti-programmed death-ligand 1 (PDL1) treatments. Magnetic resonance imaging (MRI) was used in to predict the status of MGMT, a promoter of methylation that has been related to better outcomes on GBM patients integrated with expression data (accuracy of 73%; (*Kanas et al., 2017*)).

*Fischer et al. (2018)* reported a new method for histopathological image analysis—sparse coding—using a dictionary optimised for biomedical images. They stated that they generally obtained better performance rates compared to transfer learning. In *Yu et al. (2016)*, they predicted the prognosis of non-small cell lung tumours. Using the CellProfiler software, they extracted 9,879 quantitative characteristics and trained different algorithms, such as SVM or random forest. Finally, with a variant of the SVM algorithm, they achieved an AUROC of 81%. Besides, they developed a low-complexity method for classification and disease grading in histopathological images. This method—discriminative feature-oriented dictionary learning (DFDL)—learns from specific class dictionaries in such a way that under a dispersion restriction, the learned dictionaries allows it to represent a new image in a simplified way. However, it is unable to represent samples from other classes. *Coudray et al. (2018)* used histopathology images of lung cancers to classify squamous cell carcinomas, adenocarcinomas and normal samples with a 97% of AUROC. In the work of *Cheerla & Gevaert (2019)*, they were able to extract information from several datasets and obtain a model capable to predict patient prognosis. *Ertosun & Rubin (2015)* subtyped gliomas with CNN algorithms by using raw images for this task; there was more than 90% accuracy for glioma classification and almost 80% for glioma grade identification. *Rendleman et al. (2019)* used a CNN to evaluate distinct histological tumour growth patterns such as solid, micropapillary, acinar and cribriform (84% accuracy). An important work was developed in *Janowczyk et al. (2019)*. They developed an unsupervised encoder to compress four data modalities, including whole slide images (WSIs), into a single feature vector for each patient. The model was trained with TCGA data and predict single cancer overall survival, achieving a C-index of 0.78 overall.

It is important to highlight the need to pre-process the histopathological images before their analysis. This step is crucial to achieve great performances in the models. The images housed in TCGA are not homogeneous in size, shape and brightness. Therefore, it is necessary to use a pre-processing stage in order to standardise all the images before the analysis. Open source tools as HistoQC (*Janowczyk et al., 2019*) are relevant in the extraction of knowledge and the good use of images in research.

## Biological questions solved by ML algorithms

In addition to all the existing omics data in TCGA, the inclusion of the clinical information from each patient increases the ability to generate analytical models. The dependent variable in supervised learning problems can potentially be any of the 100 clinical outcomes offered by TCGA, depending on the biological response to be answered. For classification problems, researchers have information on the anatomical division of the neoplasm, the clinical stage of the patient, TNM status, MSI status, ethnicity, age and gender, survival and/or relapse of the tumor among others. Thus, we can infer whether we can predict the anatomical division of the tumour or its clinical stage from the methylation marks of the patients (among other possibilities). For regression, we can use the initial age at diagnosis or the prognosis of the patient by means of the Karnofsky Performance Status Scale. Also, independently of clinical data, classificacion and regression models could be created to determinate other omics outcomes. For instance, sobreexpression of driver genes, methylation status or mutation types. In addition, the data storaged in TCGA repository allows any potential researcher to study survival in the cohort: it presents data on life status and the days that have elapsed between events, such as the death of the patient or other events of interest (relapse or disease-free survival).

The quantification of the number of papers published for each cancer subtype is shown in Fig. 3. As shown in this figure, the most used are those with the highest number of samples: BRCA, LUAD and OV. The great number of dimensions and observations, together with the large number of available clinical variables (pathological state, TNM classification status, drug effect, treatment response, etc.) generates an ideal data analysis environment for the use of both supervised and unsupervised ML techniques. For supervised learning problems, contingent on the dependent variable to be predicted, these problems may be regression (patient survival time, expression of a specific gene or individual age) or classification (classification of patients according to some driver gene status, disease or metastasis stages, etc.) problems. In terms of unsupervised learning problems, most work focuses on finding new subtypes of the disease. As for the other tumours, there is a significant decrease in the number of publications, mainly due to the number of samples collected. This fact is due to the intrinsic functioning of the ML algorithms, which, because they work on the basis of examples, are able to generalise more as the number of observations in their training phase increases. We can observe in the Fig. 3 also how there are several works that use different cohorts in the same analysis. After reviewing these papers, two trends have been observed in this type of article. Firstly, there are those that train models to predict cross-sectional and/or basic conditions of tumours. For example, in *Fischer et al. (2018)* they predicts MSI status from

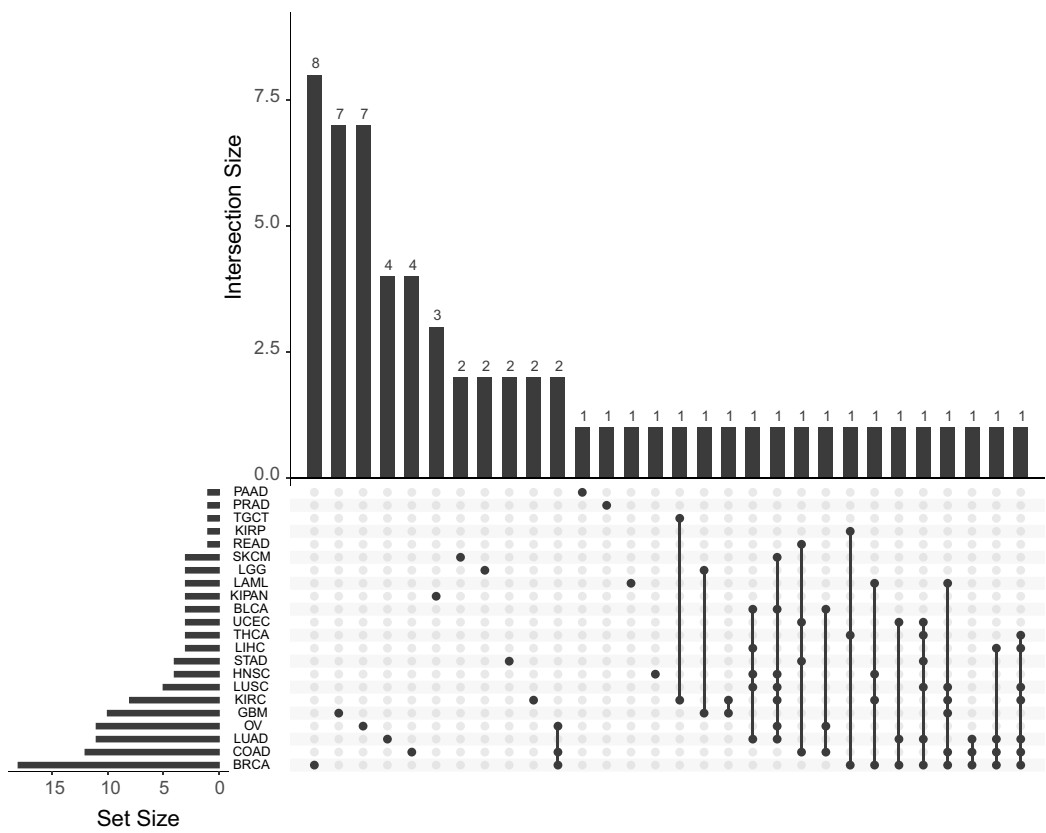

**Figure 3  Number of papers published with each of the TCGA cohorts.** Upset plot showing the number of works published with each tumor type and their combinations.

histopathological images. In this case, the different TCGA cohorts are treated together for the training of the models. On the other hand, other works have been identified in which the cohorts are used independently. These works are mainly based on model improvements or development of new technologies that are then tested with each cohort. This is the example of *Chen et al. (2018b)*, where they develop a new model of autoencoders for the search of new genetic signatures. This model is later validated in each of the available TCGA cohorts. Another example is *Cheerla & Gevaert (2017)*, where they obtain a model that recommends the type of treatment from miRNA expression data. This recommendation is validated in the different TCGA cohorts. Therefore, there are many approaches that can be used by researchers to use this type of data.

In this review, we classify the identified works on five major groups according biological problem solved. Although there are more than 100 variables in the TCGA clinical database, there is very little variability observed in the type of analysed problems.

In order to observe the distribution of publications according to this type of classification along the different types of tumours, pay attention to the Fig. 4. Figure 4 shows the distribution of the published papers according to the different types of tumors and the type of biological problem. The different biological problems show a different distribution according to tumor type. It can be seen how prognosis prediction is more

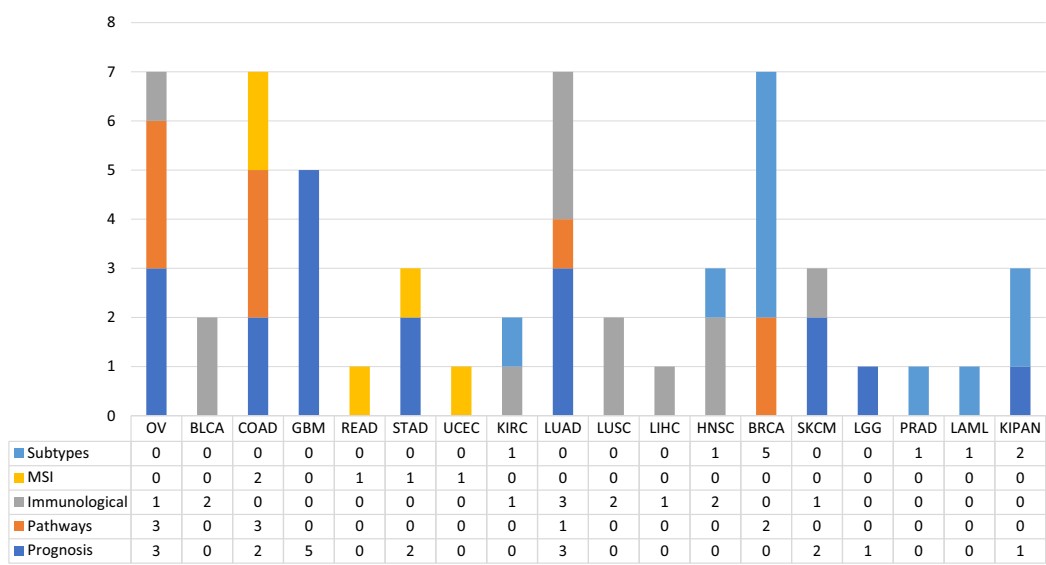

**Figure 4** The proportion of published works with ML techniques according to the type of biological problem.

common in GBM cohorts. In this case GBM is a type of tumour with high mortality rates, so it is a cohort where there are numerous events with which robust ML models can be created. Following GBM, OV and LUAD cohorts were the most used. Furthermore, it is observed how this type of problem is addressed in different cohorts. This is not the case for MSI prediction, as few tumours are defined by MSI status. The most common ones in this case are COAD, READ, STAD and UCEC. Paying attention to the prediction of subtypes, we see that the BRCA cohort is the most used. Regarding the immunological phenotype, the works have used cohorts mainly of solid tumours, which are the ones that present the best response to treatments with immunological therapies. Finally, few tumours have been addressed in the prediction of pathways. The works identified used the OV, COAD, LUAD and BRCA cohorts. The following sections are a review of the works according to the five classes identified.

### Prognosis prediction

The prognosis in the different types of cancer varies greatly due to their heterogeneity, their environment and their unique behaviour in each patient. It is therefore crucial to be able to predict the events that will develop in the patient and have a direct effect on the prognosis of the cancer. These events can be deaths, recurrence and/or relapse events, metastases or the classification of patients into specific stages. Numerous studies have been identified that have addressed this field of study with ML-based analyses.

Within this category, many of the papers identified have aimed to predict events related to patient survival time. Furthermore, it has been observed that expression data are the most used in this type of problem, due to their better performance in predictions, together with methylation data (*Stephen & Lewis, 2013*). In *Wong, Rostomily & Wong (2019)* they use them as input from a deep learning network, while in *Fatai & Gamieldien (2018)* they use the SVM algorithm. In both problems they obtained gene signatures that were

highly correlated with the survival events of the patients. Other works have addressed this type of problem by integrating expression data with other data sets. For example, in *Yasser et al. (2018)*, using FS techniques, they obtained subgroups of features from sets of ANC, methylation and expression. In *Zhang et al. (2016)*, they add a layer of complexity, adding to the integration of miRNA data by means of multiple kernel and FS techniques. This technique was also used in *Srivastava et al. (2013)* for the integration of expression and miRNA data. On the other hand, one paper has used only lncRNA data capable of predicting survival events over 19 months (*Cheng, 2018*). Works were also identified that have addressed this problem from histopathological images. In these cases, they extract characteristics from the images in order to train different types of ML algorithms and be able to predict survival times and/or events (*Ing et al., 2017*; *Yu et al., 2016*; *Powell et al., 2017*).

In addition to survival events and times, there are other events that are interesting to predict for clinical practice. In this case, the events of tumour recurrence, which are when the tumour is detected again after a treatment process. Knowing, therefore, the probabilities of a cancer relapse in a given patient is interesting for clinicians. Using the ML approach, this problem has been addressed mainly with transcriptomical expression data. For example, in *Wang et al. (2018)*, from miRNA data, lncRNA and mRNA identified 36 features capable of classifying with 91% accuracy whether a tumour will recur or not. In *Zhou et al. (2018)*, *Sun et al. (2018a)*, *Xu et al. (2017)*, similar performances were obtained with RNASeq data, while in *Wei (2018)* from RNASeq data predict metastasis processes. On the other hand, in *Feng et al. (2018)* they predicted recurrence based on data from the tumour microenvironment.

Usually the different types of tumours are classified in different stages which correlate with different prognoses. Therefore, this has been another problem addressed by the researchers, in which the ML has been able to offer a solution. Again, the RNASeq data were the most used to address this problem. In *Fan et al. (2018)*, they obtained a signature of 12 genes capable of distinguishing patients with lung cancer with different risks, while in *Chen et al. (2017)* they identified pathways of interest capable of classifying the different stages of lung adenocarcinoma. For example, in *Yang, Xu & Zeng (2018)*, they used only the features corresponding to lncRNA, obtaining a signature of six lncRNA capable of classifying patients with melanoma according to their stages.

### Immunological phenotype prediction

Currently, one of the most successful and promising therapies against cancer are drugs that act against immune checkpoint inhibitors (ICI). These drugs block the proteins produced by certain immune cells to prevent immune responses from becoming too strong. The activation of these checkpoints can cause the cells of our immune system not to be able to kill the cancer cells. The treatment of most types of tumours is helped by this type of therapy, although there are some that do not respond in the same way. This is the case of HGSOC tumours. In *Dai et al. (2018)*, they analysed genomic data from HGSOC patients to predict their immune phenotype of the tumour microenvironment. After a comparison with the analysis of other solid tumours, such as BLCA, SKCM, KIRC, LUSC

and LUAD, they identified ten dominant factors that determine the immunogenicity of HGSOCs. Using the ML they were able to classify tumours with high and low cytolytic activity, noting also that mutations in BRCA1 may be a good predictive biomarker for guiding ICI therapies of HGSOC patients.

Moreover, they developed and independently validated an eight-feature signature based on CD8 cell radiomic imaging for the response to (PD)-1 and (PD-L1). This imaging predictor provides a promising way to predict the immunologic phenotype of tumors and infer clinical outcomes for cancer patients who had been treated with anti-PD-1 and PD-L1.

### Pathways prediction

Some of the genetic drivers specific to each tumour are well known, as well as certain pathways that influence the process of tumour development. Although the identification of status is a complex issue, it holds a great deal of information in the diagnosis and treatment of patients. This is why researchers have addressed this problem using ML techniques. After the review carried out in this work, works have been detected that were able to model this problem. Most of them are based on RNASeq data, with which they infer the status of different cancer driver pathways (*Rykunov et al., 2016*), damaged pathways (*Klein, Stern & Zhao, 2017*) and level of apoptosis (*Salvucci et al., 2017*). In *Chen et al. (2012)*, RNASeq data and copy number data are used to detect pathways capable of differentiating expression patterns between different phenotypes. In the case of *Ou-Yang et al. (2017)*, they developed a cross-platform method for the identification of new molecular pathways related to tumour types.

### MSI status prediction

Microsatellite instability is the mutation predisposition of certain tumours due to defects in the DNA mismatch repair machinery. It is of great importance to identify MSI status in certain tumours as it is a great predictor and marker for diagnosis and treatment. In this review two papers were identified that have addressed this problem with MSI techniques. The first of these, called *Wang & Liang (2018)*, classified the different MSI subtypes based on mutational annotation data. They used an SVM algorithm and obtained a total accuracy of 0.91 for the COAD, READ, STAD and UCEC cohorts. They used a total of 22 features for the classification, such as the count of SNPs, indels, total mutations, missence mutations or the ratio between mutations and SNPs. On the other hand, in *Chen et al. (2018c)* they made a classification from the expression data. Using ML algorithms and FS techniques they obtained a classifier capable of discerning the different subtypes.

### Subtypes prediction

Finally, another problem that has been addressed by researchers and where ML techniques can contribute significantly is the prediction of the different subtypes of the disease. It is interesting to recognise which are the different omic data sets that hold enough information to build a classification system robust enough to obtain the appropriate yields. As usual, RNASeq was the technique par excellence from which the data were obtained

to train the models (*Yang et al., 2014*; *Graudenzi et al., 2017*; *Gao et al., 2017*). In addition, the expression data were combined with other sets such as miRNA (*Wilop et al., 2016*; *Nair et al., 2015*), methylation (*List et al., 2014*) or miRNA and methylation (*Nguyen et al., 2017*).

In addition to expression data, two papers have used exclusively image data to classify subtypes of the disease. Firstly from MRI images (*Sutton et al., 2017*) and with qCT-TA data (*Kocak et al., 2018*). Other work, for example, used mutation data (*Vural, Wang & Guda, 2016*) and miRNA data (*Muhamed Ali et al., 2018*).

It is logical to think that the ML algorithms now attempt to analyse the most studied problems to determine whether they can reach the same conclusions as conventional statistical approximations. In general, ML approximations analyse the importance of each of the variables in the dataset without making any a priori assumptions, so the generalisation of the model does not have to be based on inherent biological knowledge of the data. Although there are ML approximations that base the selection of genes from each data platform to certain pathways of interest (*Seoane et al., 2013*), this field is still open field for new approximations.

One study observed that the ML algorithms reached similar conclusions and also provided a certain degree of diversity in the results (*Liñares Blanco et al., 2019*). This outcome aids the examination of new omics variables that might be of interest to study the development of cancer. Cancer is a multifactorial and complex disease, so it makes sense that the analysis should consider the differences that characterise the patients as a whole and not individually.

### A deeper examination of the BRCA cohort

The TCGA consortium jointly analysed genomic DNA copy number arrays, DNA methylation, exome sequencing, mRNA arrays, miRNA sequencing and reverse-phase protein arrays (*The Cancer Genome Atlas Network, 2012b*). In this study, they demonstrated the existence of four main classes of breast cancer by combining data from five platforms; there was great heterogeneity. Mutations in only three genes (TP53, PIK3CA and GATA3) occurred in more than 10% of all the samples. In addition, they identified two new subgroups defined by protein expression—produced primarily by the tumour microenvironment. Besides, the comparison of basal-type breast tumours with high-grade serous ovarian tumours showed a myriad of molecular similarities, a finding that indicates a related aetiology and similar therapeutic opportunities.

In one study (*Ciriello et al., 2015*), the authors discovered that invasive lobular carcinoma (ILC) is a clinically and molecularly distinct disease. In this case, patients with ILC show CDH1 and PTEN loss, AKT activation and mutations in TBX3 and FOXA1. The proliferation and expression of genes related to the immune system defined three ILC subtypes.

The findings made by TCGA are leading the way in the search for new treatment and diagnostic opportunities for patients, in this case, with breast cancer. Although the work of the TCGA has been exhaustive, the possibilities offered by giving free access to its

data are enormous. For this reason, many researchers have taken these data as a reference and have reported results of great interest to the community.

We identified several publications that utilised ML to analyse TCGA BRCA data. There are published works using miRNA data (*Sherafatian, 2018*), methylation data (*Hao et al., 2017*), expression data (*Wen, Li & Chang, 2018*), integrative analysis of expression and methylation data (*Cappelli, Felici & Weitschek, 2018*) and even expression data from isomiRs (*Liao et al., 2018*). These works achieved prominent outcomes, notably the ability to infer that the problems of classification for diagnosis (healthy or disease patients) are problems that the ML algorithms solve quite easily, even with different types of data.

Several papers have been published to address this patient stratification. For example, to classify the subtypes of PR, ER and HER2 with miRNA data (*Sherafatian, 2018*; *Liao et al., 2018*), the status of the basal subtype through the analysis of images with deep learning algorithms (*Chidester, Do & Ma, 2018*) and the different subtypes of BRCA by the expression of molecular pathways (*Graudenzi et al., 2017*), mutation data (*Vural, Wang & Guda, 2016*) or even the integration of expression and methylation data (*List et al., 2014*). Cancer subtypes can be studied by unsupervised learning techniques and the integration of different data (expression, methylation, miRNA and CNV) (*Nguyen et al., 2017*).

Finally, other works have studied the interaction between miRNA and mRNA (*Koo, Zhang & Chaterji, 2018*; *Ghoshal et al., 2018*), the identification of altered pathways by mRNA expression data (*Klein, Stern & Zhao, 2017*) or by integrating expression and mutation data (*Rykunov et al., 2016*), the response to drugs in different cell lines (*Daemen et al., 2013*; *Geeleher et al., 2017*) and the identification of variants by means of genomic data (*Dong et al., 2016*) and by means of images with artificial vision techniques (*Sutton et al., 2017*).

## CONCLUSIONS

Many studies on cancer have been performed in recent years with ML that uses molecular data. These data have mainly included diagnostic studies, prognosis or patient stratification. More recently, there have been promising results in response to drugs or genetic interactions. In this review, we investigated and identified those relevant works that have used TCGA data through algorithms or pipelines of analysis based on ML.

ML techniques can extract the underlying knowledge from a set of data, so it is relevant to understand the appropriateness of the data. In other words, these techniques must be used with certain precautions. Indeed, researchers should be aware that the conclusions they obtain may be biased due to poor data selection or analytical methodology. Among the different learning techniques, supervised learning has analysed the most problems using TCGA data. This endeavour has emphasised the use of genetic expression data through different variants of the SVM algorithm. There are still infinite opportunities and possibilities for the exploitation of TCGA data with ML. ML techniques can reach conclusions that are similar to conventional approaches and also to obtain a degree of variability that is extremely useful when searching for novel predictors.

It is clear that we are still at an early stage in the analysis of this pathology and it is necessary to develop and use more complex algorithms. For example, the use of

kernel-based models can integrate different datasets in the same process. The integration of data in the analysis of complex and multifactorial diseases continues to be a challenge for which it is necessary to invest even more time and money in finding better algorithms. As discussed above, the quantity of existing data will not stop growing and all derive from the same biological sample. Thus, it is expected that the connection between omics platforms can improve the performance of the models. It is still necessary to take a step forward in the development of multidimensional ML models for cancer research.

Complex problems, such as the prediction of different cell statuses (methylation, apoptosis or mutation), are already being tackled with promising results. We and others hope that the links between biological information extracted from the same patient will be further explored in order to elucidate the origin of the disease by ML techniques. Currently, the focus is on certain types and subtypes of cancer (e.g., BRCA, LUAD or OV), usually due to the number of people afflicted with it and the importance attributed by society. It is also necessary to increase investment in the generation of data that is related to relatively minor or especially aggressive cancer types in order to provide the algorithms with sufficient information in their learning phase and to avoid biases in their learning.

In this work, we exhaustively reviewed studies that have used ML techniques for the analysis of different types of cancer using TCGA data. In our opinion, the era of individual analysis has passed and we are entering the era of data integration studies—at the clinical-genomic level as well as medical imaging or evolution analysis by means of time series. We are working on the development of complex data integration algorithms in different fields, one of which is artificial intelligence. There are currently ML models that are demonstrating great effectiveness and are gaining followers. These methods include the aforementioned deep learning techniques, but research is required to render the results understandable and explain why a certain prediction is made, especially from a clinical point of view. The great challenge of the integration techniques is the incessant increase in the number of dimensions and the heterogeneity of the data sets generated from the same patient/biological process (*Kristensen et al., 2014*).

Finally, we hope that this review will serve as a starting point for researchers in bioinformatics and computer science who are interested in studying cancer, as well as those researchers who are more focused on the use of ML techniques to know the potential of their algorithms with TCGA data. More research and the development of new algorithms are required to overcome the disease.

## ABBREVIATIONS

| | |
|---|---|
| **ML** | Machine Learning |
| **TCGA** | The Cancer Genome Atlas |
| **MSI** | Microsatellite Instability |
| **BRCA** | Breast Cancer Adenocarcinoma |
| **GBM** | Glioblastoma Multiforme |
| **SNP** | single nucleotide polymorphisms |
| **LUSC** | Lung squamous cell carcinoma |
| **OV** | Ovarian Cancer |

| SVM | Support Vector Machines |
| RF | Random Forest |
| WGS | Whole Genome Sequencing |
| CNN | Convolutional Neural Networks |
| TIL | Tumour-infiltrating lymphocytes |
| MRI | Magnetic resonance imaging |

## ACKNOWLEDGEMENTS

We thank Dr. Jose A. Seoane for comments during the preparation of this review.

### Funding

This work was supported by the "Collaborative Project in Genomic Data Integration (CICLOGEN)" PI17/01826 funded by the Carlos III Health Institute from the Spanish National plan for Scientific and Technical Research and Innovation 2013–2016 and the European Regional Development Funds (FEDER)—"A way to build Europe." and the General Directorate of Culture, Education and University Management of Xunta de Galicia (Ref. ED431D 2017/16), the "Galician Network for Colorectal Cancer Research" (Ref. ED431D 2017/23) and Competitive Reference Groups (Ref. ED431C 2018/49). CITIC, as Research Center accredited by Galician University System, is funded by "Consellería de Cultura, Educación e Universidades from Xunta de Galicia", supported in an 80% through ERDF Funds, ERDF Operational Programme Galicia 2014–2020, and the remaining 20% by "Secretaría Xeral de Universidades" (Grant ED431G 2019/01). The funders had no role in study design, data collection and analysis, decision to publish, or preparation of the manuscript.

### Grant Disclosures

The following grant information was disclosed by the authors:
Carlos III Health Institute: PI17/01826.
European Regional Development Funds (FEDER).
General Directorate of Culture, Education and University Management of Xunta de Galicia: ED431D 2017/16.
Galician Network for Colorectal Cancer Research: ED431D 2017/23.
Competitive Reference Groups: ED431C 2018/49.
Consellería de Cultura, Educación e Universidades from Xunta de Galicia.
Secretaría Xeral de Universidades: ED431G 2019/01.

### Competing Interests

The authors declare that they have no competing interests.

## Author Contributions

- Jose Liñares-Blanco conceived and designed the experiments, performed the experiments, analyzed the data, prepared figures and/or tables, and approved the final draft.
- Alejandro Pazos conceived and designed the experiments, authored or reviewed drafts of the paper, and approved the final draft.
- Carlos Fernandez-Lozano conceived and designed the experiments, analyzed the data, authored or reviewed drafts of the paper, and approved the final draft.

## Data Availability

This research is a Literature Review; there are no raw data or code files.

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
