# Peer review of "Machine learning analysis of TCGA cancer data"

_PeerJ Computer Science, doi:10.7717/peerj-cs.584_

## Round 0.1 · original submission · Minor Revisions

Please address the reviewers' comments and provide point-by-point response.

Reviewer 1 ·

Basic reporting

The authors present a review to reflect the state of the art in machine learning particularly on The Cancer Genome Atlas Program data.
First, some of the work of the TCGA consortium itself is described to give the reader a good foundation. However, the main goal of this work is to identify and discuss those works that have used the TCGA data to train different ML approaches. The authors classify them according to three main criteria: the type of tumor, the type of algorithm, and the predicted biological problem. One of the conclusions drawn in this work shows a high density of studies based on two main algorithms: Random Forest and Support Vector Machines and naturally an increase is described from the use of deep artificial neural networks. An increase of integrative models of multi-omic data analysis is also presented. Biological problems were classified into five types: Prognosis prediction, tumor subtypes, microsatellite instabilities, immunological aspects, and specific signaling pathways.
A clear trend was found in the prediction of these conditions according to tumor type. This is why a greater number of papers have focused on the BRCA cohort, while specific papers for survival, for example, have focused on the GBM cohort, due to its largeNumber of events.

Any work dedicated to fighting cancer is important. This work is a very good contribution and very useful for the international research community and it is very well written, well motivated and good to read. For all these reasons, this reviewer endorses this work, recommends acceptance, and makes some recommendations below to help further improve the work:

1) For the beginner in this field, a list of all abbreviations used would be very helpful, e.g. MSI = Microsatellite Instable is not mentioned anywhere and newcomers should be able to find their way quickly.

2) Figure 2, the descriptions are very hard to read - maybe the image can be optimized

3) Figure 2 b - is practically illegible

4) Figure 4 also very hard to read

5) In the summary section or before, a bit of an outlook on future important research topics should be given, e.g. comprehensibility, interpretability to further explore causal relationships which is eminently important in cancer research. Here, however, it is totally important to point out that in the medical field always several different components contribute to a result - but this is often negated in current machine learning, consequently, it would be good to say so, and to point to a brand new current work that deals exactly with this matter [x].
[x] Holzinger, A., Malle, B., Saranti, A. & Pfeifer, B. 2021. Towards Multi-Modal Causability with Graph Neural Networks enabling Information Fusion for explainable AI. Information Fusion, 71, (7), 28-37, doi:10.1016/j.inffus.2021.01.008

6) Readers within the subsection "A general perspective of unsupervised learning with TCGA data" may also interested be interested in this brand new work [y]:
[y] Jean-Quartier, C., et al. A. 2021. Mutation-based clustering and classification analysis reveals distinctive age groups and age-related biomarkers for glioma. BMC medical informatics and decision making, 21, (77), 1-14, doi:10.1186/s12911-021-01420-1

Experimental design

nothing to add - very good

Validity of the findings

nothing to add - very good

Additional comments

please see comments above.

Reviewer 2 ·

Basic reporting

No comment

Experimental design

No comment

Validity of the findings

No comment

Additional comments

Summary of Paper:
In this paper, the authors provide a comprehensive review on literature using machine learning techniques for the analysis of different types of cancer using TCGA data. Starting with explaining the methodologies used in this survey, the authors present the main results obtained by the TCGA consortium followed by reviewing the capabilities of machine learning algorithms to solve biological problems. Overall, the paper is well written. Below are my minor concerns on the paper.

Minor Concerns:
• On line 102, 'Machine learning as a source o new knowledge' -> 'Machine learning as a source of new knowledge.'
• On line 107, ‘To his aim …’ -> ‘To this aim’.
• It is advisable to write the article inclusion/exclusion criteria on lines 127-136 more clearly. For example, I am assuming that the last point ‘Article/conference manuscripts using machine learning marginally or without solid biological conclusions’ should be the exclusion criteria.
• ‘neuron networks’ on lines 241 and line 244 should be replaced by ‘neural networks’.
• Consider rephrasing the sentence ‘In this Figure 4 is interesting to observe how the different problems addressed are distributed differently in the types of tumour.’ on lines 391-392.
• ‘Immnulogical phenotype prediction’ -> ‘Immunological phenotype prediction’ on line 441
• ‘immunologicc phenotype’ -> ‘immunologic phenotype’ on line 455.

---

## Round 0.2 · accepted · Accept

There is no further comment from the reviewer.

Reviewer 2 ·

Basic reporting

no comment

Experimental design

no comment

Validity of the findings

no comment

Additional comments

The authors have addressed all my previous comments.